# Therapeutic Effect of *Rumex japonicus* Houtt. on DNCB-Induced Atopic Dermatitis-Like Skin Lesions in Balb/c Mice and Human Keratinocyte HaCaT Cells

**DOI:** 10.3390/nu11030573

**Published:** 2019-03-07

**Authors:** Hye Ryeon Yang, Hyunkyoung Lee, Jong-Hyun Kim, Il-Hwa Hong, Du Hyeon Hwang, Il Rae Rho, Gon Sup Kim, Euikyung Kim, Changkeun Kang

**Affiliations:** 1College of Veterinary Medicine, Gyeongsang National University, Jinju 52828, Korea; 2015210922@gnu.ac.kr (H.R.Y.); leehy@gnu.ac.kr (H.L.); jkim@gnu.ac.kr (J.-H.K.); ihhong@gnu.ac.kr (I.-H.H.); pooh9922@hanmail.net (D.H.H.); gonskim@gnu.ac.kr (G.S.K.); ekim@gnu.ac.kr (E.K.); 2Institutes of Animal Medicine, Gyeongsang National University, Jinju 52828, Korea; 3Institutes of Agriculture and Life Science, Gyeongsang National University, Jinju 52828, Korea; irno12@gnu.ac.kr

**Keywords:** *Rumex japonicus* Houtt., atopic dermatitis, DNCB, skin lesion, MAPK, NF-κB, TNF-α

## Abstract

*Rumex japonicus* Houtt. (RJ) is traditionally used in folk medicines to treat patients suffering from skin disease in Korea and other parts of East Asia. However, the beneficial effect of RJ extract on atopic dermatitis (AD) has not been thoroughly examined. Therefore, this study aimed to investigate the anti-inflammatory effects of RJ on AD in vitro and in vivo. Treatment with RJ inhibited the phosphorylation of mitogen-activated protein kinase (MAPK) as well as the activation of nuclear factor-kappa B (NF-κB) in tumor necrosis factor-α (TNF-α) stimulated in HaCaT cells. The five-week-old Balb/c mice were used as an AD-like mouse model by treating them with 1-chloro-2, 4-dinitrobenzene (DNCB). Topical administration of RJ to DNCB-treated mice significantly reduced clinical dermatitis severity, epidermal thickness, and decreased mast cell and eosinophil infiltration into skin and ear tissue. These results suggest that RJ inhibits the development of AD-like skin lesions by regulating the skin inflammation responses in HaCaT cells and Balb/c mice. Thus, RJ may be a potential therapeutic agent for AD.

## 1. Introduction

Atopic dermatitis (AD) is a multifactorial skin disease, with complex interactions. Various factors, including immunological abnormalities, contribute to the pathogenesis and development of AD [1]. The early onset of AD in infancy results in it being the most prevalent chronic skin disorder in childhood and can affect individuals throughout their lifetimes [2]. The common symptoms of AD include dry, inflamed skin, intense pruritis, itching and skin hypersensitivity. In some instances, AD can also cause recurring rashes, persistent scratching, erythematous plaques, and small bumps like blisters that may leak extracellular fluid. In chronic severe cases, AD causes sleep disturbance which may leads to insomnia, psychological and emotional distress, and low quality of life [3,4,5].

The current treatment for AD involves topical steroids, emollients and oral anti-histamines as the first-line therapy, but many patients are still worried about the long term use of these agents, and side-effects are frequently observed [2,6,7,8]. As a result, the importance of developing novel therapeutic agents for AD has increased recently. Natural products, such as astaxanthin and red ginseng, are increasingly being implicated in the regulation of inflammatory cytokines and chemokines in the development of AD-like skin lesions [9,10]. 

*Rumex japonicus* Houtt. (RJ) is a perennial herb that is distributed throughout Japan, Korea, and China. It contains a large number of anthraquinones, flavonoids and triterpenoids, and has potential clinical applications in skin disease due to its antioxidant activity [11]. We previously reported that RJ had a hair growth-promoting effect via mitogen-activated protein kinases (MAPKs) and Wnt/β-catenin pathways in human keratinocytes (HaCaT) and mice [12]. Although RJ has been showed to have pharmacological activity in hair growth, the effect of RJ as an anti-inflammatory agent for AD remains poorly understood. This study was designed to assess the anti-AD effects of RJ on TNF-α-induced immune responses in HaCaT cells and a 1-chloro-2, 4-dinitrobenzene (DNCB) application mouse model. 

## 2. Materials and Methods

### 2.1. Materials

Bovine serum albumin (BSA), Dulbecco’s modified Eagle’s Medium (DMEM), Fetal bovine serum (FBS), penicillin, streptomycin and trypsin were purchased from Gibco-BRL (Grand Island, NY, USA). Dimethyl sulfoxide (DMSO) and 3-(4,5-dimethylthiazol-2-yl)-2,5 diphenyltetrazolium bromide (MTT) were obtained from Sigma-Aldrich Inc. (St. Louis, MO, USA). Antibodies for phospho-Akt (Ser473), phospho-ERK, phospho-p38, phosphor-IκBα, NF-κB p65, Lamin B1 and GAPDH were obtained from Cell Signaling Technology (Beverly, MA, USA). TNF-α was purchased from R&D Systems (Tokyo, Japan). All other reagents used were of the purest grade available.

### 2.2. Preparation of *Rumex japonicus* Houtt Extract

The dried roots of *Rumex japonicus* Houtt. (RJ) were provided by Keratin Korea (Busan, Korea). The roots were washed 3 times with tap water to remove impurities and dried at room temperature, then stored at 4 °C until use. Dried roots were ground into powder with a grinder and extracted 3 times with 95% ethanol at 25 °C for 3 days, after which the extract was filtered through the Advantech No. 3 filter paper (Cole-Parmer, Osaka, Japan). The filtered liquid was evaporated using a rotary vacuum evaporator (Tokyo Rikakikai Co., Ltd., Tokyo, Japan). The final step was lyophilization under vacuum to dryness.

### 2.3. Cell Culture and Cell Viability

Cell viability was measured by MTT assay. Briefly, human keratinocyte cells (HaCaT) were maintained in DMEM with 10% FBS and 100 μg/mL penicillin-streptomycin at 37 °C in a 5% CO_2_ humidified incubator. HaCaT cells were plated at a density of 4 × 10^4^ cells/well in 24-well plates and cultured overnight in growth DMEM media. Cells were removed by gentle washing with fresh culture medium and treated with various concentrations of RJ and incubation for 24, 48 and 72 h. MTT solution (5 mg/mL) was then added to each well and incubated for an additional 3 h at 37 °C. Finally, DMSO was added to solubilize the formazan salt and the amount of formazan generated was determined by measuring the optical density (OD) at 540 nm using a GENios^®^ microplate spectrophotometer (PowerWaveTMXS, BioTek Instruments, Inc., Winooski, VT, USA). 

### 2.4. Western Blot Analysis

Cells were incubated at a density of 5 × 10^4^ cells/well in 6-well plates for 24 h in complete DMEM. After the adaptation process, cells were treated with RJ for 30 min, and then stimulated with TNF-α (10 ng/mL) for 30 min in serum-free culture medium. After cells had been washed with cold PBS, the treated cells were collected by scraping with 300 μL of RIPA buffer (TransLab, Daejeon, Korea) containing protease inhibitor. Lysates were separated using 10% SDS-polyacrylamide gel and then transferred to PVDF membranes (Bio-Rad, Hercules, CA, USA). Western blot were probed with specific primary antibodies overnight at 4 °C. Following incubation with horseradish peroxidase-conjugated secondary antibody (Bethyl Laboratories Inc., Montgomery, AL, USA) for 1 h at room temperature, the blots were visualized using an enhanced chemiluminescence method (ECL, Amersham Biosciences, Buckinghamshire, UK) and analyzed using ChemiDoc XRS (Bio-Rad, CA, USA). Densitometry analysis was performed with a Hewlett-Packard scanner and Image Lab software.

### 2.5. Experimental Animals

Five-week-old female Balb/c mice were purchased from Samtako Inc. (Osan, Korea) and cared for in Gyeongsang National University laboratory animal research center. Mice were housed in cages under standard conditions with food/water available ad libitum and a 12 h light/dark cycle. The room temperature and humidity were 23 ± 2 °C and 35–60%, respectively. The animal study protocol used in this work was approved by the Institutional Animal Care and Use Committee of Gyeongsang National University, and the animal study protocol number was GNU-160623-M0015.

### 2.6. Experimental Study with RJ for AD

A total of 25 mice were randomly divided into five groups (*n* = 5 per group): group 1, vehicle-treated group (Sham); group 2, vehicle with DNCB (Control); group 3 and 4, 4 mg/mL and 8 mg/mL RJ with DNCB, respectively; group 5, dexamethasone with DNCB group (positive control). The doses of RJ were selected considering that there was no clinical pathology in our preliminary toxicity study (data not shown). Atopic dermatitis was induced in the mice by treatment with 1-chloro, 2,4-dinitrochlorobenzene (DNCB). On day 0, the hair on the dorsal skin of the mice in all groups was shaved using an electric razor. During days 1–3, 200 μL of 0.5% DNCB solution (dissolved in a 3:1 mixture of acetone and olive oil) was applied once daily to the dorsal skin and ear of the mice for three days in the control, RJ, and dexamethasone groups. The sham group received a vehicle treatment (only acetone/olive oil). After this initial sensitization treatment, RJ was applied topically to the skin and ears of the mice every day for 3 weeks. The positive control of dexamethasone treatment was applied through a 1.5 mg/kg I.P injection three times per week. Animals were sacrificed at experimental day 35. The exact experimental schedule is illustrated in Figure 1.

### 2.7. Measurement of Ear Thickness and Organ Weight

Ear thickness was measured with a micrometer on the day of sacrifice. The micrometer was applied near the tip of the ear just distal to the cartilaginous ridges, and the thickness was recorded in micrometers. Weights of lymph nodes and spleen were measured with an electronic balance.

### 2.8. Histopathological Studies

The skin and ear lesions were sliced and tissue slices were fixed in 10% buffered-neutral formalin for 24 h. The fixed tissue slices were embedded in paraffin wax, sectioned, deparaffinized, and rehydrated using standard techniques. Sections 5 μm thick were subjected to hematoxylin and eosin (H&E) staining and toluidine blue staining for the detection of various inflammatory cells. Histopathological changes were examined by light microscopy. An arbitrary scope was given to each microscopic field viewed at a magnification of 100 (skin) and 200 (ear).

### 2.9. Statistical Analysis

The results are expressed as mean ± standard deviation (S.D.). One-way analysis of variance (ANOVA) was used to evaluate the significance of difference between the two mean values. * *p* < 0.05 and ** *p* < 0.01 were considered to be statistically significant.

## 3. Results

### 3.1. The Optimal Treatment Concentration of RJ Extract on HaCaT Cells

To determine the optimal treatment concentration of RJ on HaCaT cells, an MTT assay was performed. The RJ extract had no cytotoxic effect at a concentration of 50 μg/mL for 24 h, but as shown in Figure 2A, the viability of cells treated with an RJ concentration of 1600 μg/mL declined by about 44% after 24 h (IC_50_ approximately 1150.72 μg/mL). We also measured the effect of RJ in the presence or absence of TNF-α. RJ had no cytotoxic effect at concentrations of up to 50 μg/mL with 10 ng/mL TNF-α stimulation in the HaCaT cells (Figure 2B). Therefore, we used 25 and 50 μg/mL of RJ with 10 ng/mL of TNF-α in our subsequent experiment.

### 3.2. RJ Treatment Modulates TNF-α Induced Changes in MAPK Family and NF-κB Pathway of Proteins in HaCaT Cells

To investigate the molecular mechanism of the anti-inflammatory activity of RJ in TNF-α-stimulated keratinocytes, we first examined whether RJ inhibits the activation of mitogen-activated protein kinase (MAPK) and Akt signaling pathways. HaCaT cells were pretreated with RJ (25 and 50 μg/mL) for 30 min, followed by induction by TNF-α for 30 min. We analyzed phosphorylation of ERK 1/2, p38, and Akt by Western blot. Dose-dependent RJ treatment inhibited the phosphorylation of p38, ERK 1/2 and Akt in TNF-α-stimulated cells (Figure 3A,B). Next, we examined the activation of NF-κB and IκBα degradation in TNF-α-stimulated HaCaT cells. In an unstimulated state, NF-κB was retained in the cytosol as an inactive complex with its inhibitory protein IκBα, which blocks the nuclear importing sequences of NF-κB. The transcriptional activation of NF-κB was regulated by the phosphorylation of its functionally active subunit p65/RelA with serine 536 residue in its transcriptional activation domain [13]. The dose-dependent RJ treatment inhibited the phosphorylation of IκBα and translocation of NF-κB p65 (Figure 3C). Thus, these results indicate that RJ had an anti-inflammatory effect on the accumulation of allergic modulators by blocking MAPK, Akt and suppressing NF-κB activation.

### 3.3. RJ Suppressed Clinical Severity of AD Skin Symptoms in DNCB-Induced Balb/c Mice

To assess the effectiveness of RJ against AD-like skin lesions, Balb/c mice were treated by DNCB to cause AD-like skin lesions, followed by treatment with RJ and dexamethasone, as described in Figure 1. As expected, the application of DNCB led to significant inflammation. As shown in Figure 4A, the repeated topical application of DNCB significantly increased ear thickness in the DNCB-treated group compared with the sham group. In addition, the RJ-treated group reduced ear thickness in a dose-dependent manner during the whole study period. Since AD often develops as a systemic immune response, it can affect the immune organs [14,15,16,17]. Therefore, we evaluated that weight of the lymph nodes and spleen was measured in the final week to examine whether the topical application of RJ provided an anti-AD effect in mice. The control group results showed an elevation of lymph node and spleen weight compared with the sham group. The RJ treatment group showed a dose-dependent inhibition of lymph node weight (Figure 4B). In addition, the spleen weight was slightly decreased after RJ treatment (Figure 4C). Thus, these results show that the topical application of RJ can have anti-AD effects.

### 3.4. Effect of RJ on AD-Like Skin Lesion in Mice

AD increased skin thickness and excessive lymphocyte infiltration into the dermis. The skin and ear was stained with H&E and toluidine blue to examine whether RJ improves AD-like skin pathogenesis. H&E staining revealed that the control group drastically increased the thickness of both the epidermal and dermal tissues, whereas the RJ treatment group significantly ameliorated these changes compared with the control group, especially for epidermal tissue (Figure 5A). Moreover, toluidine blue staining indicated a prominent number of mast cells in the dermal area, whereas the RJ treatment group saw a dose-dependent decrease in mast cell numbers (Figure 5B). Thus, these findings contribute to a better understanding of the pathogenesis of AD-like skin lesion in mice.

## 4. Discussion

The biological properties and health benefits of *Rumex japonicus* Houtt. (RJ) have been studied intensively over the past three decades on a variety of actions, including anti-oxidant, anti-microbial, anti-tumor, anti-inflammatory and anti-allergic effects [12,18,19,20]. The curative effects of RJ on skin disease and its complications seems to be related to its inhibition of T-helper 2 cell response [21]. Although the anti-allergic effects of RJ have already been reported, the mechanistic understanding of its pharmacological roles for atopic dermatitis is poorly elucidated. The principal finding of the present study is that RJ regulates the inhibition of MAPKs and NF-κB activation in TNF-α induced HaCaT cells and attenuates the development of DNCB-induced atopic dermatitis lesions in Balb/c mice.

Keratinocytes such as HaCaT cells play pivotal roles in skin disease and are commonly used for the in vitro testing of anti-inflammatory skin drugs [22]. Stimulation of keratinocytes by TNF-α leads to activation of various signaling pathways that involve MAPKs, Akt, and NF-κB [23]. One of the most extensively investigated intracellular signaling cascades involved in pro-inflammatory responses are the MAPK pathways [24]. MAPK pathways fall into several different subgroups, including ERK1/2, p38 and JNK. They are important signal transducers for cell survival and can regulate several cellular processes including proliferation, differentiation, survival and apoptosis. MAPKs are also involved in the inflammatory cytokine network. ERK1/2 belongs to the MAPK family and is crucial in the control of cell growth, cell differentiation and cell survival, and p38 acts as a key component in the induction of inflammation and immune system disorders [25]. We conducted experiments to explore the effects of RJ on the TNF-α-induced expression of MAPK in HaCaT cells. Treatment of TNF-α significantly enhanced the phosphorylation of ERK and p38 expression. In contrast, treatment with RJ significantly inhibited the expression of ERK and p38 signal in TNF-α-stimulated HaCaT cells.

Akt activity increases throughout the entire process of skin disease [26] and can activate the transcription factor NF-κB [27]. It has been suggested that the Akt pathway contributes to the activation of NF-κB through the phosphorylation of IκBα [27]. Recent findings indicate that TNF-α can activate NF-κB, which is responsible for the expression of pro-inflammatory cytokines [28]. Upon this cytokine stimulus, the IκB proteins are phosphorylated and degraded, which allows NF-κB to translocate into the nucleus where it can bind to specific promoter regions of target genes and activate the expression of inflammatory cytokine genes. Thus, the inhibition of NF-κB activation has been suggested as an anti-inflammatory strategy in AD [1,28]. As shown in our data, Akt activity was significantly reduced, preventing phosphorylation through treatment with RJ. Furthermore, RJ inhibited the degradation of IκBα and the nuclear localization of NF-κB.

Mouse models of AD allow the in-depth investigation of pathogenesis for allergic skin inflammation. Since the description of the Nc/Nga mouse as the first AD model in 1997, a number of mouse models have been developed [29]. From these mouse models, we selected the DNCB-induced mouse model due to its simplicity and histological characteristics. Repeated topical application of DNCB on the skin of mice induced many of the histopathological symptoms of AD within a month. Experimental data show that the AD-like skin lesions induced by DNCB treatment were predominantly controlled by cellular immune responses [30,31,32]. We evaluated the effects of RJ skin severity and histopathological changes in DNCB-induced mice. Topical treatment with RJ for 3 weeks showed a dose-dependent and effective improvement in skin symptoms and reduced the thickness of dorsal skin and ear skin thickness in DNCB-induced mice. We also found that topical RJ treatment reduced lymph node and spleen weight after these increased as a result of DNCB-sensitization, which may be involved in immune activity. In addition, the topical administration of RJ attenuated the infiltration of inflammatory eosinophils and mast cells in the ears and skin. From these results, we confirmed the therapeutic effects of RJ in AD-like skin lesions. In addition, various immune cells have been identified in relation to the development of AD.

In conclusion, the topical administration of RJ inhibits the development of DNCB-induced atopic dermatitis skin lesions in Balb/c mice. In addition, the results of the in vitro study showed that RJ suppressed the activation of ERK, p38, Akt and NF-κB in TNF-α-stimulated HaCaT cells. Taken together, these results suggest that the topical application of RJ may be an effective alternative therapy for the management of AD. Further studies are needed to address the serum IgE, histamine, inflammatory cytokine and chemokine levels, and determine which components of RJ contribute to its efficacy in the treatment of inflammatory skin disease.

## Figures and Tables

**Figure 1 nutrients-11-00573-f001:**
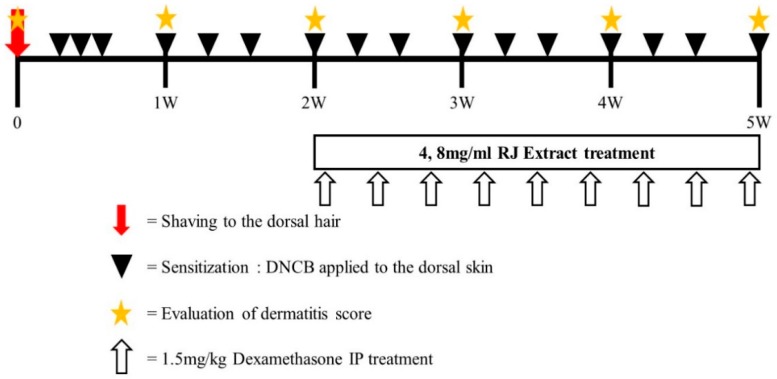
Experimental schedule for the induction of atopic dermatitis (AD) lesions. Mice were divided into five groups (*n* = 5 per group). To induce AD-like immunological and skin lesions, DNCB was applied to the dorsal skin and ears.

**Figure 2 nutrients-11-00573-f002:**
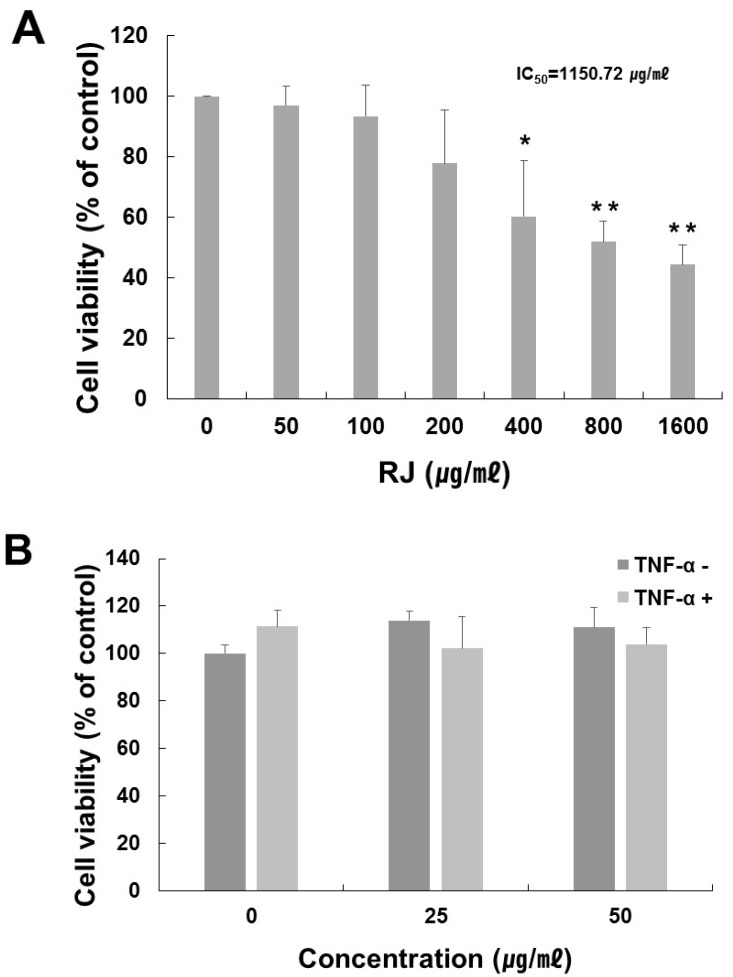
Cell viability of various treatment concentration of RJ on HaCaT cells. (**A**) HaCaT cells were treated with RJ at the indicated concentrations for 24 h. (**B**) Cell viability of RJ (0, 25, 50 μg/mL) with or without TNF-α (10 ng/mL) in HaCaT cell. HaCaT cells were pretreated with 0, 25 and 50 µg/mL RJ for 30 min then stimulated with 10 ng/mL of TNF-α for 30 min. Cell viability was then determined by MTT assay. The data shown are the mean ± SD of three independent experiments. Significant difference from control group, * *p* < 0.05, ** *p* < 0.01.

**Figure 3 nutrients-11-00573-f003:**
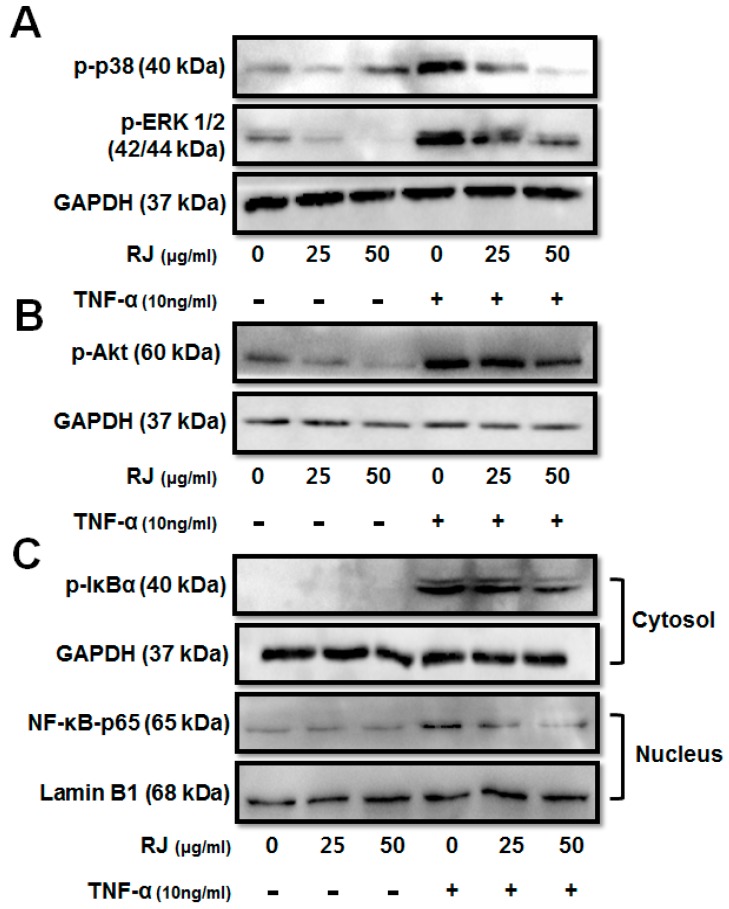
The inhibitory effect of RJ on TNF-α-stimulated mitogen-activated protein kinases (MAPK) and nuclear factor-kappa B pathways. HaCaT cells were pretreated with 0, 25 and 50 µg/mL RJ for 30 min then stimulated with 10 ng/mL of TNF-α for 30 min. (**A**) MAPKs including ERK1/2, p38 were assessed by Western blot analysis. (**B**) Akt phosphorylation was inhibited in a dose-dependent manner after TNF-α stimulation of the HaCaT cells. (**C**) Inhibition of NF-κB pathway in TNF-α stimulated HaCaT cells was assessed by Western blot analysis. The data shown are the mean ± SD of three independent experiments.

**Figure 4 nutrients-11-00573-f004:**
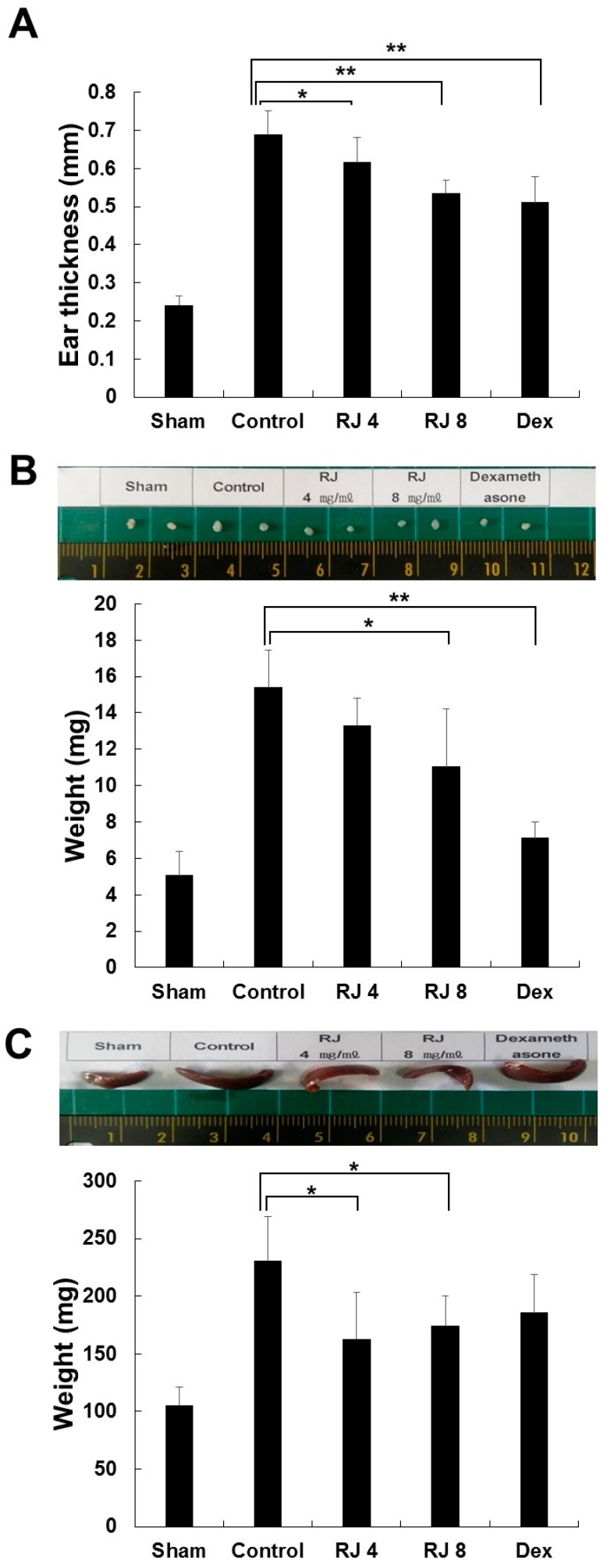
Inhibitory effects of RJ on DNCB-induced AD skin symptoms in Balb/c mice. The mice were divided into five groups: Vehicle (Sham), DNCB + vehicle (Control), DNCB + RJ 4 mg/mL (RJ 4), DNCB + RJ 8 mg/mL (RJ 8), positive control (DEX). (**A**) Ear thickness was measured with a micrometer. Five mice were used per group. (**B**,**C**) Organ sizes were compared by photographic images. Organ and whole-body weight of five mice per group were measured. Five mice were used per group. The data shown are the mean ± SD of three independent experiments. Significant difference from control group, * *p* < 0.05, ** *p* < 0.01.

**Figure 5 nutrients-11-00573-f005:**
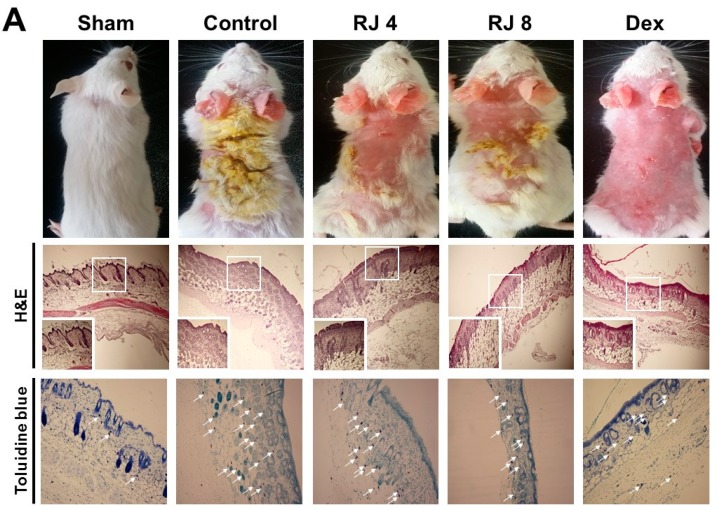
Effect of RJ on histological changes of the dorsal skin and ear lesions. (**A**,**B**) Skin and ear lesions were removed and fixed in 10% formaldehyde solution. Skin sections were cut and stained with hematoxylin and eosin and toluidine blue. The immune cells (arrows) are indicated. Photographs were taken under a regular light microscope at a magnification of 100× (skin), 200× (ear). Five mice were used per group.

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
