# Peer review of "Therapeutic Effect of *Rumex japonicus* Houtt. on DNCB-Induced Atopic Dermatitis-Like Skin Lesions in Balb/c Mice and Human Keratinocyte HaCaT Cells"

_nutrients, 2019, doi:10.3390/nu11030573_

Round 1

Reviewer 1 Report

The experiment is well designed and the topic is of interest.  Furthrmore, there is the need  of new therapeutical options for the treatment  of AD. I would like the authors address some minor issues, as following:

- Line 175-176: Since AD often develops as a systemic immune response, it can affect the immune organs [11-13]. The importante of the gut-skin axis is well recognized and the gut is an important “immune organ” , Can they give a brief comment about? I suggest to add an appropriate reference (eg paper by ONeill) 

      -    Lines 248-249: We also found that topical RJ treatment reduced lymph node and spleen weight 248 that were increased by DNCB-sensitization, which may be involved in immune activity. Did the authors also look at the gut or not?

- Lines 41-42:  

 It contains a larger number of anthraquinones, flavonoids and triterpenoids and has potential clinical applications in skin disease due to its antioxidant activity. Redarding the falvonoids content, culd the authors give some more details about the flavonoids concentrations, if known?  

- Lines 256-257: I suggest may be instead of “to be”, given the preliminary data

References: A part of reference n 3, the ref n 2 and 4 refer to animal model . 

As the sentence correctly refer to the fact that “many patients are still worried about long term use of these agents ..” I suggest the authors to include review discussing the benefits and the potential risks of topical therapyes, eg emollients, oral anti-histamines topical steroids etc  (see and cite the review by D’Auria et al APJAI 2016 and Chia BK, Tey HL, Dermatitis 2015). 

Issue regarding quality of life (see Line 33). Given the importance of the issue I suggest to add references specifically focusing on it (CL Carrol Pediatric Dermatology 2005; Boccardi D Minerva Pediatrica 2017)

Author Response

Response to reviewers:

To reviewer 1: Thank you very much for your valuable comments. According to your comments, we have changed manuscript as follow. In the manuscript, revised portions are shown in red color.

Point 1. Line 175-176: Since AD often develops as a systemic immune response, it can affect the immune organs [11-13]. The importance of the gut-skin axis is well recognized and the gut is an important “immune organ”, Can they give a brief comment about? I suggest to add an appropriate reference (eg paper by ONeill)

-Answer: Thank you for the suggestion. We have added Dr. Catherine A. O’Neill paper in our revised manuscript.

Point 2. Lines 248-249: We also found that topical RJ treatment reduced lymph node and spleen weight 248 that were increased by DNCB-sensitization, which may be involved in immune activity. Did the authors also look at the gut or not?

-Answer: Thank you for the question. In this study, we treated with topical administration on skin and ear. So, we did not check the change of gut after RJ treatment. If we study another experiment with oral treatment, we will check the change of gut as you suggested.

Point 3. Lines 41-42: It contains a larger number of anthraquinones, flavonoids and triterpenoids and has potential clinical applications in skin disease due to its antioxidant activity. Redarding the falvonoids content, culd the authors give some more details about the flavonoids concentrations, if known?

-Answer: Thank you for the question. Actually, we have checked the antioxidant activity of RJ using total polyphenol content assay. In our preliminary data, 95% EtOH extraction of RJ (158.3 ug/mg) has shown higher total polyphenol content than 80% MeOH extraction of RJ (113.5 ug/mg). So we selected 95% EtOH extraction of RJ for current experiment.

Point 4. Lines 256-257: I suggest may be instead of “to be”, given the preliminary data

-Answer: Thank you for the suggestion. We have corrected sentences in line 256-257 as you suggested

Point 5. References: A part of reference n 3, the ref n 2 and 4 refer to animal model.

As the sentence correctly refer to the fact that “many patients are still worried about long term use of these agents” I suggest the authors to include review discussing the benefits and the potential risks of topical therapyes, eg emollients, oral anti-histamines topical steroids etc (see and cite the review by D’Auria et al APJAI 2016 and Chia BK, Tey HL, Dermatitis 2015).

Issue regarding quality of life (see Line 33). Given the importance of the issue I suggest to add references specifically focusing on it (CL Carrol Pediatric Dermatology 2005; Boccardi D Minerva Pediatrica 2017)

-Answer: Thank you for the suggestion. We have added several papers in our revised manuscript as you suggested.

Reviewer 2 Report

The present study by Yang et al addresses an interesting topic, i.e. the therapeutic effect of Rumex japonicus on atopic dermatitis (or in this case an AD-like inflammation mouse model). 

In order to approach this question, the authors performed to sets of experiments, on the one hand on HaCaT cells (in vitro), on the other hand in an AD-like mouse model (in vivo). The experiments are well described and performed with proper control groups, there are however some major questions that remain open.

Major issues: 

1. In the in vitro experiments, the effect of RJ on cell viability was assessed. A negative effect of RJ on cell viability was found when using concentrations of 400ug/ml and higher. For the further in cvitro experiments, the authors thus used 25 or 50ug/ml.  In the mouse experiments however, concentrations of 4 or 8mg/ml were used, which (according to the in vitro data) would be toxic. 

Why did the authors use these concentrations? Would they be affect cell viability in vivo / was there any toxicitiy? Was there any effect on inflammation in the mouse model when using concentrations of 25 or 50ug/ml (or the other way around: what happens when using 4 or 8mg/ml for the in vitro experiments)?

2. When assessing the effect of RJ on AD-like skin inflammation, it would be most interesting to further address/show parameters reflecting cutaneous inflammation, e.g.:

- How did RJ (as compared to the other conditions) affect the inflammatory infiltrate (i.e. the different cell types such as T cells, eosinophils, dendritic cells, etc. in skin?

- What was the effect of RJ on the molecular inflammatory signature (expression of IL17, IL22, IL4, …) in the inflamed mouse skin?

Also the expression of antimicrobial peptides (which normally decreased in AD-skin) would be an interesting parameter to assess

- Did you perform measurements at other time points (in addition to day 35)? Did you have similar results?

3. Although TNF-alpha is a major mediator of skin inflammation, it is not the most important one in AD (role of Th2 cytokines, IL22, etc.). The link between the in vitro assays and AD thus is not clear and does not seem an ideal approach to the question. A more detailed explanation of this approach might be helpful.

Minor issues:

1. There are some typos in the manuscript (eg p 5 line 156: treat instead of treated; p2 line 63 filterd).

Author Response

Response to reviewers:

To reviewer 2: Thank you very much for your valuable comments. According to your comments, we have changed manuscript as follow. In the manuscript, revised portions are shown in red color.

Point 1. In the in vitro experiments, the effect of RJ on cell viability was assessed. A negative effect of RJ on cell viability was found when using concentrations of 400ug/ml and higher. For the further in vitro experiments, the authors thus used 25 or 50ug/ml. In the mouse experiments however, concentrations of 4 or 8mg/ml were used, which (according to the in vitro data) would be toxic.

Why did the authors use these concentrations? Would they be affect cell viability in vivo / was there any toxicitiy? Was there any effect on inflammation in the mouse model when using concentrations of 25 or 50ug/ml (or the other way around: what happens when using 4 or 8mg/ml for the in vitro experiments)?

-Answer: Thank you for the question. The ranged from 25~50 ug/ml concentrations had no cytotoxic effect in HaCaT cells, so we determined those concentrations for our in vitro study. Also we chosen 4 and 8 mg/ml concentrations of RJ because our previous study reported positive pharmacological in vivo effect without side effect.

1) Reference 1: Lee H et al.,The Hair Growth-Promoting Effect of Rumex Japonicus Houtt. Extract. Evid Based. Complement. Alternat Med. 2016, 1873746.

Point 2.  When assessing the effect of RJ on AD-like skin inflammation, it would be most interesting to further address/show parameters reflecting cutaneous inflammation, e.g.:

- How did RJ (as compared to the other conditions) affect the inflammatory infiltrate (i.e. the different cell types such as T cells, eosinophils, dendritic cells, etc. in skin?

-Answer: Thank you for the question. In our histopathological findings, we only found that RJ reduced epidermal thickness and mast cell infiltration both in dorsal skin and ear tissue.

- What was the effect of RJ on the molecular inflammatory signature (expression of IL17, IL22, IL4,) in the inflamed mouse skin?

Also the expression of antimicrobial peptides (which normally decreased in AD-skin) would be an interesting parameter to assess

-Answer: Thank you for the comment and question. We have mentioned the limitation of RJ treatment on AD in our conclusion part. We did not check serum immunoglobulin E (IgE), inflammatory cytokine levels and expression of antimicrobial peptides in this experiment. Therefore, we need further investigation and improvements in the near future.

- Did you perform measurements at other time points (in addition to day 35)? Did you have similar results?

-Answer: Thank you for the question. We did not perform this experiment over 5 weeks. Moreover, we did not find any negative effect in RJ treated mice group during whole experiment period and modified other researchers report.

1) Reference 1: Ok S et al. Effects of Angelica gigas Nakai as an Anti-Inflammatory Agent in In Vitro and In Vivo Atopic Dermatitis Models. Evid Based Complement Alternat Med; 2018: 2450712

Point 3. Although TNF-alpha is a major mediator of skin inflammation, it is not the most important one in AD (role of Th2 cytokines, IL22, etc.). The link between the in vitro assays and AD thus is not clear and does not seem an ideal approach to the question. A more detailed explanation of this approach might be helpful.

-Answer: Thank you for the question. TNF-alpha induces some proinflammatory cytokines including IL-1beta, IL-6, IL-8, and itself by activation of NF-kappa B or MAPKs (p38, JNK, ERK). And dysregulated NF-kappa activity is related with inflammatory response like AD and is important in the physiology and pathology of skin. Moreover, Th2 type chemokine can be induced by TNF-alpha in keratinocytes. Other researchers also used TNF-alpha induced keratinocyte model for studying AD. So we determined to use TNF-alpha induced keratinocyte model for our study.

1) Reference 1: Cho JW et al., Curcumin attenuates the expression of IL-1beta, IL-6, and TNF-alpha as well as cyclin E in TNF-alpha-treated HaCaT cells; NF-kappaB and MAPKs as potential upstream targets. Int J Mol Med. 2007;19(3):469-74.

2) Reference 2: Park H et al., Ramalin Isolated from Ramalina Terebrata Attenuates Atopic Dermatitis-like Skin Lesions in Balb/c Mice and Cutaneous Immune Responses in Keratinocytes and Mast Cells. Phytother Res. 2016 12:1978-1987

3) Reference 3: Cho K et al., Pyrus ussuriensis Maxim. leaves extract ameliorates DNCB-induced atopic dermatitis-like symptoms in NC/Nga mice Phytomedicine, 2018 48, 7683

Point 4. There are some typos in the manuscript (eg p 5 line 156: treat instead of treated; p2 line 63 filterd).

- Answer: Thank you for the comment. We have found it a typing error and corrected it.

Round 2

Reviewer 2 Report

none